# Impedimetric Immunosensor Utilizing Polyaniline/Gold Nanocomposite-Modified Screen-Printed Electrodes for Early Detection of Chronic Kidney Disease

**DOI:** 10.3390/s19183990

**Published:** 2019-09-16

**Authors:** Muhammad Omar Shaikh, Boyanagunta Srikanth, Pei-Yu Zhu, Cheng-Hsin Chuang

**Affiliations:** 1Institute of Medical Science and Technology, National Sun Yat-sen University, Kaohsiung 80424, Taiwan; omar.offgridsolutions@gmail.com; 2Department of Mechanical Engineering, Southern Taiwan University of Science and Technology, Tainan 71005, Taiwan; srikanthrockz5121@gmail.com (B.S.); 4a10h008@stust.edu.tw (P.-Y.Z.)

**Keywords:** chronic kidney disease, immunosensor, electrochemical impedance spectroscopy, PANI/AuNCs nanocomposite

## Abstract

The presence of small amounts of human serum albumin (HSA) in urine or microalbuminuria (30–300 µg/mL) is a valuable clinical biomarker for the early detection of chronic kidney disease (CKD). Herein, we report on the development of an inexpensive and disposable immunosensor for the sensitive, specific, and label-free detection of HSA using electrochemical impedance spectroscopy (EIS). We have utilized a simple one-step screen-printing protocol to fabricate the carbon-based three-electrode system on flexible plastic substrates. To enable efficient antibody immobilization and improved sensitivity, the carbon working electrode was sequentially modified with electropolymerized polyaniline (PANI) and electrodeposited gold nanocrystals (AuNCs). The PANI matrix serves as an interconnected nanostructured scaffold for homogeneous distribution of AuNCs and the resulting PANI/AuNCs nanocomposite synergically improved the immunosensor response. The PANI/AuNCs-modified working electrode surface was characterized using scanning electron microscopy (SEM) and the electrochemical response at each step was analyzed using EIS in a ferri/ferrocyanide redox probe solution. The normalized impedance variation during immunosensing increased linearly with HSA concentration in the range of 3–300 µg/mL and a highly repeatable response was observed for each concentration. Furthermore, the immunosensor displayed high specificity when tested using spiked sample solutions containing different concentrations of actin protein and J82 cell lysate (a complex fluid containing a multitude of interfering proteins). Consequently, these experimental results confirm the feasibility of the proposed immunosensor for early diagnosis and prognosis of CKD at the point of care.

## 1. Introduction

Chronic kidney disease (CKD) is recognized as a global health problem with growing incidence and prevalent rates of end stage renal disease (ESRD) [1,2]. The Global Burden of Disease (GBD) study estimated that 1.2 million deaths, 19 million disability-adjusted life-years, and 18 million years of life lost from cardiovascular diseases were directly attributable to reduced glomerular filtration rates, which progressively reduce with the advancement of CKD [3]. CKD occurs in stages due to a gradual loss in kidney function and is often ignored in its early stage owing to its asymptomatic nature. Screening for microalbuminuria is a valuable independent biomarker for the onset of chronic kidney as well as cardiovascular diseases in both diabetic and non-diabetic patients [4,5]. Early detection of microalbuminuria is crucial as kidney deterioration can be delayed or even prevented through effective treatment at this stage. Since regular screening of microalbuminuria is highly recommended, especially for patients with a higher risk profile such as those suffering from diabetes or hypertension, there is an urgent need for effective point of care (POC) tests that provide rapid and quantitative detection of microalbuminuria with acceptable accuracy and specificity.

Several research efforts have been aimed at developing biosensors that facilitate POC diagnostics, especially for resource-limited settings, since they are relatively inexpensive, portable, easy to use, and can deliver real-time remote healthcare monitoring [6,7,8,9]. Electrochemical biosensors are particularly attractive since they enable direct conversion of a biological event to an electrical signal and can be easily integrated with miniaturized electronics, thus making them highly suitable for quantitative detection of both catalytic and affinity bio recognition events at the POC [10]. Electrochemical immunosensors are a class of affinity-based electrochemical biosensors that report the highly selective binding of an antibody to its corresponding antigen by measuring the change in current/voltage or the perturbation thereof [11,12]. Among interrogation schemes utilized in electrochemical immunosensors, electrochemical impedance spectroscopy (EIS) is an effective and non-destructive technique for probing the antibody–antigen interaction [13,14]. When the target antigen is captured by the receptor-modified electrode surface, it alters the electrode/electrolyte interfacial properties (capacitance and charge transfer resistance) that can be analyzed using EIS. This enables direct and real-time monitoring without the need for additional labeling and amplification steps [15,16]. 

Electrochemical immunosensors for POC testing require portability and disposability and consequently utilize miniaturized electrodes that are printed on plastic or paper substrates using low-cost techniques such as screen-printing [17,18,19]. To improve antibody immobilization, electrochemical response, sensitivity, and stability, further surface modification of the working electrode is often required. Presently, a range of nanostructured modifications such as metal nanoparticles [20], conducting polymers [21], and organic–inorganic nanocomposites [22] are being researched owing to their high surface areas, facile charge transfer, and improved protein loading capabilities. Conducting polymers such as polyaniline (PANI) [23,24], polypyrrole (PPy) [25,26], and poly (3,4-ethylenedioxythiophene) (PEDOT) [27,28] are attractive as they have been shown to be a suitable immobilization matrix for a range of bio receptors including enzymes, proteins, and whole cells and can easily form nanostructured films on the electrode surface using electropolymerization or electrophoretic deposition [29]. Furthermore, they also represent a suitable matrix for the dispersion of metallic nanoparticles, resulting in novel hybrid surfaces that improve the electrochemical immunosensing performance [30]. In particular, gold nanomaterials have been widely utilized in electrochemical immunosensors owing to their biocompatibility, stability, conductivity, and chemical functionality [31,32]. 

Herein, we report on the development of a disposable electrochemical immunosensor utilizing screen-printed carbon electrodes (SPCEs) on flexible polyethylene terephthalate (PET) substrates for sensitive and specific detection of human serum albumin (HSA). To improve charge transfer and electrochemical response, we have modified the surface of the carbon working electrode with a dense and uniform PANI/gold nanocrystals (AuNCs) hybrid nanocomposite film consisting of a nanostructured polyaniline (PANI) matrix obtained by electropolymerization of aniline monomer in acidic media followed by the electrodeposition of gold nanocrystals (AuNCs). Each surface modification of the carbon working electrode was analyzed using EIS in the presence of a redox probe solution. SEM analysis was performed to study the morphology of the PANI/AuNCs nanocomposite film and ensure a uniform and stable deposition on the working electrode surface. The immunosensing feasibility of the proposed SPCE/PANI/AuNCs/Ab-HSA (anti-human serum albumin antibody) immunoelectrode was tested using spiked buffer solutions containing different concentrations of HSA protein ranging from 3 to 300 µg/mL. Finally, interference studies were performed using different concentrations of actin protein and cell lysate to confirm feasibility for specific HSA detection.

## 2. Materials and Methods

### 2.1. Chemicals and Reagents

Aniline (ACS reagent, ≥99.5%), chloroauric acid (HAuCl_4_.aq), potassium ferricyanide (K_3_[Fe(CN)_6_]), potassium ferrocyanide (K_4_[Fe(CN)_6_].3H_2_O), sulfuric acid (98%, H_2_SO_4_), sodium acetate (C_2_H_3_NaO_2_), and sodium metaperiodate (NaIO_4_) were purchased from Sigma Aldrich and were used without further purification. The Fe(CN)_6_^3^^−/4^^−^ redox probe was prepared using a solution of 0.1 M phosphate buffer saline (PBS) containing a mixture of 5 mM potassium ferro/ferricyanide. 0.1 M PBS was prepared by mixing 0.1 M sodium phosphate dibasic (Na_2_HPO_4_), 0.1 M sodium dihydrogen phosphate (NaH_2_PO_4_) and 0.1 M potassium (KCl) stock solutions with the pH adjusted at 7. Commercial screen-printing carbon paste (C-1011-6) was obtained from Advanced Electronic Materials Inc., Taiwan. Mouse monoclonal anti-human serum albumin antibody (Ab-HSA), native human serum albumin (HSA), native actin protein and bovine serum albumin (BSA) were purchased from Abcam. The J82 bladder cancer cell line was obtained from the Bioresource Collection and Research Center (BCRC), Hsinchu, Taiwan. The cells were cultured in McCoy’s 5A medium (GIBCO, Thermo Fisher Scientific) followed by lysing and harvesting using a protein extraction buffer (GE Healthcare). The original protein concentration in the obtained lysate was determined using a Bio-Rad DC protein assay kit. 

### 2.2. Apparatus

The SPCE was fabricated using a commercial screen printer (ATMA CHAMP Ent. Corp., Taoyuan, Taiwan). A tabletop electrochemical station (PGSTAT100N, Metrohm Autolab, Utrecht, Netherlands) was used for electropolymerization of PANI and electrodeposition of AuNCs. The same electrochemical station was used to perform EIS measurements for each step during electrode modification and after immunosensing. The surface morphology of the SPCE before and after surface modification with PANI/AuNCs nanocomposite film was characterized using a field emission scanning electron microscope (Hitachi SU-8010, Tokyo, Japan) operated at an accelerating voltage of 15 kV.

### 2.3. Fabrication of SPCE

The SPCE was fabricated using single-step screen-printing of the carbon paste on flexible and transparent PET substrates and the process is simple, low cost, and scalable. A schematic of the carbon-based three-electrode system, comprising of a working (Ø = 4 mm), counter, and reference electrode, is shown in the image in Figure 1a, and an image of the obtained SPCE is presented in Figure 1b. After screen-printing, the SPCE was cured in a vacuum oven at 120 °C for 30 min to improve the mechanical and electrical properties and ensure a strong adhesion to the PET substrate.

In this study, we have utilized a three-electrode configuration with the inclusion of a reference electrode which allows potential changes of the working electrode to be measured independent of changes that may occur at the counter electrode, thus reducing background effects. Alternatively, certain two-electrode configurations such as interdigitated microelectrodes have also been widely used to fabricate impedance-based biosensors due to their inherent simplicity, low ohmic drop, and ease of miniaturization. In our previous study, we have demonstrated the feasibility of utilizing screen-printed interdigitated microelectrodes for dielectrophoretic trapping of antibody-conjugated nanoprobes and impedance-based immunosensing [32]. Furthermore, we have also performed numerical simulations of the electric fields generated when an alternating voltage is applied to the interdigitated microelectrodes [33]. However, since the electrode gap width significantly influences the electric field strength and sensitivity of impedance biosensors, a major drawback of printing as compared to conventional thin-film fabrication techniques such as photolithography is the minimum achievable microelectrode gap width, which is limited to about 25–50 µm. 

### 2.4. Surface Modification with PANI/AuNCs 

The working electrode surface was sequentially modified with PANI and AuNCs to obtain the PANI/AuNCs hybrid nanocomposite film. A conventional glass electrochemical cell was utilized consisting of the screen-printed working electrode and a separate platinum wire and Ag/AgCl as the counter and reference electrodes, respectively. This enables electropolymerization of PANI and electrodeposition of the AuNCs selectively on the working electrode surface. The systematic surface modification of the SPCE and immunosensor operation is schematically illustrated in Figure 2. Aniline was electropolymerized on the electrode surface from an aqueous solution containing 0.5 M H_2_SO_4_ and 0.1 M aniline monomer. Electropolymerization was performed using cyclic voltammetry (CV) in a potential range of −0.2–0.8 V at a scan rate 100 mV/s for 50 successive cycles. After a washing step using deionized water, the AuNCs were electrodeposited from an aqueous solution containing 5 mM HAuCl_4_ precursor and 0.1 M KNO_3_ using linear sweep voltammetry (LSV) from −1.5 to 0 V at a scan rate of 10 mV/s. After a washing step using deionized water, the working electrode was ready for antibody conjugation. 

### 2.5. Antibody Conjugation and Blocking Step

Before conjugation, the antibody was oxidized in an aqueous solution of 1 mM sodium metaperiodate and 0.1 M sodium acetate with the solution pH maintained at 5.5. During this step, the hydroxyl groups (–OH) in carbohydrate moieties of the antibodies are oxidized to aldehyde groups (–CHO), which can covalently link to terminal amino groups (–NH_2_) present on the PANI surface via the formation of a peptide bond. Then, 100 µL of Ab-HSA (0.3 mg/mL) was dropped on the modified SPCE surface and incubated in a dark moisture-saturated environment for 3 h at room temperature. This was followed by washing with 0.1 M PBS to remove any unbound antibodies. Next, the electrode surface was incubated with BSA solution (1% w/w in 0.1 M PBS) for 30 min at room temperature to block any active sites that will result in non-specific protein absorption. After a washing step was performed to remove any unbound BSA proteins, the fabricated SPCE/PANI/AuNCs/Ab-HSA immunoelectrodes were stored in 0.1 M PBS at 4 °C until ready for immunosensing. 

### 2.6. Immunosensor Operation

The fabricated SPCE/PANI/AuNCs/Ab-HSA immunoelectrodes were tested using two sets of sample solutions: (i) has-spiked 0.1 M PBS solution in a concentration range of 3–300 µg/mL for testing feasibility of HSA detection. (ii) Actin protein- and J82 cell lysate-spiked 0.1 M PBS solution with concentrations of 75, 150, and 300 µg/mL for testing the specificity of the immunosensor. Briefly, 100 µL of sample solutions were dropped on the sensing area to perform immunosensing for 30 min. Next, a washing step was performed to remove any unbound proteins. All electrochemical measurements, including bare SPCE, after surface modifications and after immunosensing, were made by dropping 100 µL of 5 mM Fe(CN)_6_^3^^−^^/4^^−^ redox probe solution on the sensing area and analyzing the corresponding impedance spectra (Bode and Nyquist plots). EIS measurements were made by applying a small-amplitude sinusoidal voltage of 10 mV in a frequency range of 0.1–100 Hz. The impedance change during immunosensing (ΔZ) was calculated at an optimized frequency of 1 Hz for which we observe the maximum stable response. Ten immunosensors were tested for each concentration to ensure repeatability. We tried to eliminate the slight differences in initial impedance of different SPCEs by normalizing the impedance response during immunosensing (ΔZ) with the baseline impedance observed after antibody immobilization and blocking at that particular electrode (Z_0_), and this is referred to as the normalized impedance variation (ΔZ/Z_0_).

## 3. Results and Discussion

### 3.1. SEM Characterization of PANI/AuNCs Modification

SEM imaging was utilized to confirm the sequential deposition of PANI and AuNCs on the bare SPCE as shown in Figure 3. It was observed from the SEM images in Figure 3b that the PANI formed a uniform, interconnected, and porous film with nanostructured morphology on the SPCE surface. The morphology of the PANI film depends on several parameters, which include aniline concentration, voltage range, scan rate, and number of scan cycles. Herein, we have shown the effects of varying the number of scan cycles on the PANI film morphology while all other parameters were kept constant. The top and cross-sectional view of the electropolymerized PANI film using 25 and 50 scan cycles is shown in the SEM images in Figure 3c,d, respectively. It was observed that the PANI film obtained using 25 scan cycles was relatively less dense with increased porosity as compared to the film obtained using 50 scan cycles. Furthermore, it was observed from the cross-sectional SEM images that the average thickness of the deposited PANI film increased from about 20 to 60 µm as the number of scan cycles increased from 25 to 50, respectively. Scan cycles over 50 resulted in a thicker polyaniline layer that was more prone to being scratched or peeled from the electrode surface, thus resulting in lower reproducibility. Consequently, optimization of individual electropolymerization parameters is critical to ensure a stable PANI coverage with suitable nanostructured morphology. The conductive PANI film acts as an extension of the SPCE and provides increased surface area for effective electrodeposition of AuNCs. As seen in the SEM images in Figure 3e,f, a dense and uniform coverage of AuNCs was obtained on the nanostructured PANI film. Furthermore, AuNCs were also deposited on the bare carbon electrode surface between the ridges and pores of the PANI film, thus ensuring that the PANI/AuNCs composite layer coated the entire working electrode surface. The presence of terminal amine groups of PANI ensures a firm deposition of AuNCs via the formation of an Au–N bond, and the electrodeposited AuNCs were not removed during washing steps. The stable PANI/AuNCs nanocomposite layer can enhance the immunosensor performance by improving the antibody-loading capability and observed electrochemical response during immunosensing. 

### 3.2. EIS Analysis

The systematic surface modification of the SPCE and observed immunosensor response was analyzed using EIS in the presence of a Fe(CN)_6_^3^^−^^/4^^−^ redox probe. EIS represents a well-established technique for characterizing an electrode/solution interface by monitoring changes in impedance over a wide frequency range. To enable a deeper understanding of the governing phenomena during impedance measurements, equivalent circuit models could be built to fit the data to a proposed model. However, even the best electrode–solution interface models do not perfectly fit the raw data or else require several fitting parameters. In some cases, the raw impedance data can be fitted to a model and changes in the model elements are reported as the sensor output. Other times, as in this study, the absolute impedance value at a particular frequency can be used. The two circuit elements most commonly utilized to analyze impedance changes include charge transfer resistance (R_ct_) and interfacial capacitance (C_int_) for faradaic and non-faradaic immunosensors, respectively. These circuit elements are often connected in parallel since the total current flowing across the electrode is the sum of the individual contributions originating from faradaic and non-faradaic processes. Faradaic EIS immunosensors, as proposed in this study, generally utilize redox active species such as ferro/ferricyanide [34] and ruthenium hexaammine [35] to monitor the kinetics of charge transfer to and from the electrode surface. The interaction of the redox species with the electrode surface determines the observed R_ct_ value (diameter of the semicircle in the Nyquist plot) and changes for each surface modification step. 

The EIS spectra, presented as Nyquist and Bode plots, are shown for each step in Figure 4a,b, respectively. The imaginary impedance component (Z”) is plotted against the real impedance component (Z’) at each excitation frequency in the Nyquist plot while the Bode plot displays the absolute impedance values observed over a range of excitation frequencies. The Nyquist and Bode plots presented in this study were found to be similar for repeated scans of a particular surface but only changed after each successive surface modification of the SPCE. It was observed from the Nyquist plot in Figure 4a that the R_ct_ value had a near 180-fold reduction after the electropolymerization step, and this can be attributed to the improved redox activity and conductivity of PANI. While the R_ct_ value was relatively similar after electrodeposition of the AuNCs, it increased after antibody immobilization and antigen binding during immunosensing. This increase can be attributed to the insulating nature of proteins that hinder the charge transfer from the redox probe to the electrode surface. Furthermore, electrostatic repulsion between the negatively charged protein surface and negatively charged redox species can also result in increased R_ct_. The successive surface modification of the SPCE was also analyzed using the Bode plots presented in Figure 4b. It was observed that the impedance change for each step dominated in the low frequency range (0.1–100 Hz). This can be expected in faradaic immunosensors, where although both real (R_ct_) and imaginary (C_int_) components change with each surface modification, the change in the real component dominates the total impedance change which is generally observed at lower frequencies. To optimize the measurement frequency, we have calculated the normalized impedance variation observed at four different frequencies (0.1, 1, 10, and 100 Hz) for immunosensing with the 300 µg/mL HSA-spiked sample solution as shown in Figure 4 c. Although the highest variation was at 0.1 Hz, the noise-to-signal ratio was large at very low frequencies and consequently, we chose a frequency of 1 Hz at which we observed the highest stable normalized impedance response during immunosensing.

### 3.3. Synergic Effect of PANI/AuNCs on Immunosensor Response

The synergic effect of the PANI/AuNCs nanocomposite film was investigated by observing the immunosensor response (normalized impedance variation) for the same HSA concentration (75 µg/mL) when the SPCE surface was modified with PANI/AuNCs nanocomposite as compared to only PANI as shown in Figure 5. Three separate electrodes were used to perform immunosensing for each electrode surface modification scheme to ensure repeatability. The normalized impedance variation during immunosensing for the SPCE/PANI/AuNCs/Ab-HSA immunoelectrode was about 24%, which is more than three times higher than that observed for the SPCE/PANI/Ab-HSA immunoelectrode. This enhanced impedance response could be attributed to improved antibody immobilization and loading capacity of the PANI/AuNCs nanocomposite layer. This may be due to a combined effect arising from (i) antibody binding to AuNCs due to the strong affinity of the antibodies to the high-energy surface of the AuNCs and (ii) covalent conjugation of oxidized antibodies to available terminal amino groups of PANI via the formation of an amide bond. Furthermore, the strong interactions between the PANI and AuNCs could improve the electron transfer rate of proteins while providing a biocompatible and favorable microenvironment for immunosensing. 

### 3.4. Feasibility for HSA Detection

The feasibility of the proposed SPCE/PANI/AuNCs/Ab-HSA immunoelectrode was tested by performing immunosensing using HSA-spiked 0.1 M PBS sample solutions where the concentration of HSA was varied from 3 to 300 µg/mL. This concentration range was utilized since it covers the clinically relevant range of microalbuminuria (30–300 µg/mL) and can, thus, enable early detection of CKD. The normalized impedance variation observed during immunosensing displays a linear relationship with the concentration of HSA present in sample solution as shown in Figure 6a, with a linear regression equation Y = 0.10359X + 5.10409 and coefficient of determination (R^2^) value of about 0.96. The normalized impedance variations for HSA concentrations of 3, 30, 75, 150, and 300 µg/mL were calculated to be 7.66% ± 0.42%, 14.95% ± 1.68%, 23.94% ± 2.41%, 31.22% ± 2.82%, and 51.32% ± 7.16%, respectively, where each concentration was tested using 10 immunosensors. These preliminary results confirm the feasibility of the immunosensor for detection of HSA with high repeatability over the complete required clinical range. 

### 3.5. Specificity 

To confirm specificity, the SPCE/PANI/AuNCs/Ab-HSA immunoelectrode was tested using actin protein and J82 cell lysate (bladder carcinoma cell line that contains a multitude of proteins, estimated to be about 10^4^/cell [36], with some being overexpressed in the urine of individuals suffering from bladder cancer). Interfering studies were performed using spiked 0.1 M PBS sample solutions with actin and lysate protein concentrations of 75, 150, and 300 µg/mL. Three separate electrodes were used to perform immunosensing for each concentration to confirm repeatability. A clear distinction was observed between the normalized impedance variations during immunosensing with HSA and the other proteins as shown in Figure 6b. A maximum normalized impedance variation of 3.6% ± 0.6% and 4.7% ± 1% was observed for immunosensing with 300 µg/mL cell lysate and actin protein, respectively, which is significantly lower than that observed for the same concentration of HSA. Consequently, the proposed immunosensor shows high specificity for HSA detection. 

## 4. Conclusions

In summary, we have developed a disposable and label-free impedimetric immunosensor utilizing PANI/AuNCs hybrid nanocomposite modified SPCEs for sensitive and specific detection of HSA. The electropolymerization parameters were optimized to obtain a uniform and nanostructured PANI film on the carbon working electrode surface that acts as a suitable matrix for dense and even dispersion of AuNCs via electrodeposition. The PANI/AuNCs surface modification significantly improved the charge transfer capability of the printed carbon electrode and resulted in an enhanced electrochemical response during immunosensing. The immunosensor response (normalized impedance variation) increased linearly with the concentration of HSA over the complete range required for clinical detection of microalbuminuria. The preliminary results presented herein highlight the potential of the proposed immunosensor for early diagnosis and prognosis of CKD. Our future work involves testing the clinical feasibility of the immunosensor for microalbuminuria detection in human urine samples. Furthermore, by utilizing a low-cost and hand-held readout impedance module, as developed in our previous work [15], the immunosensor response can be analyzed and the data can be uploaded for cloud computing, thus enabling real-time POC detection and improved public healthcare monitoring.

## Figures and Tables

**Figure 1 sensors-19-03990-f001:**
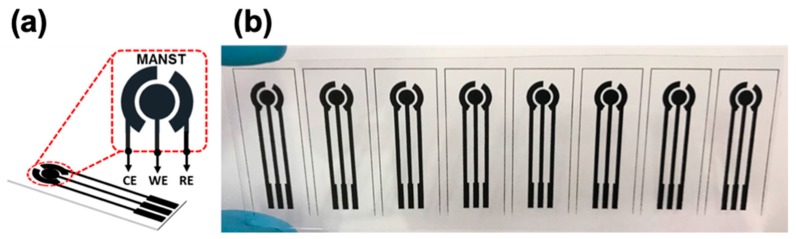
(**a**) Schematic of the carbon-based three-electrode system showing the working (WE), counter (CE), and reference (RE) electrodes. (**b**) Photographic image of an obtained batch of screen-printed carbon electrodes (SPCEs) on a flexible PET substrate.

**Figure 2 sensors-19-03990-f002:**
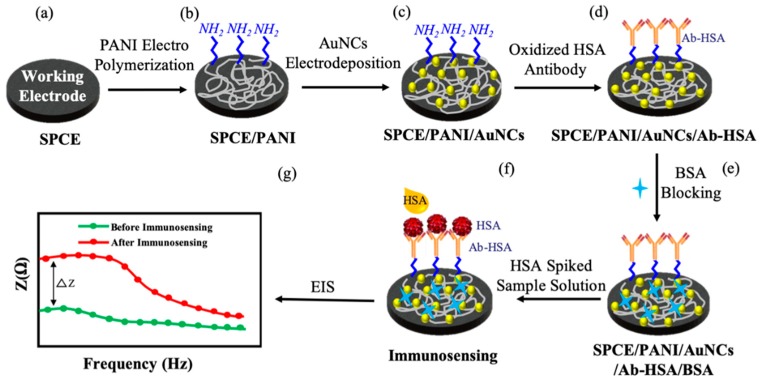
Schematic of the systematic protocol for SPCE surface modification and immunosensing. (PANI—polyaniline; AuNCs—gold nanocrystals; HSA—human serum albumin; Ab-HSA—anti-human serum albumin antibody; BSA—bovine serum albumin; EIS—electrochemical impedance spectroscopy).

**Figure 3 sensors-19-03990-f003:**
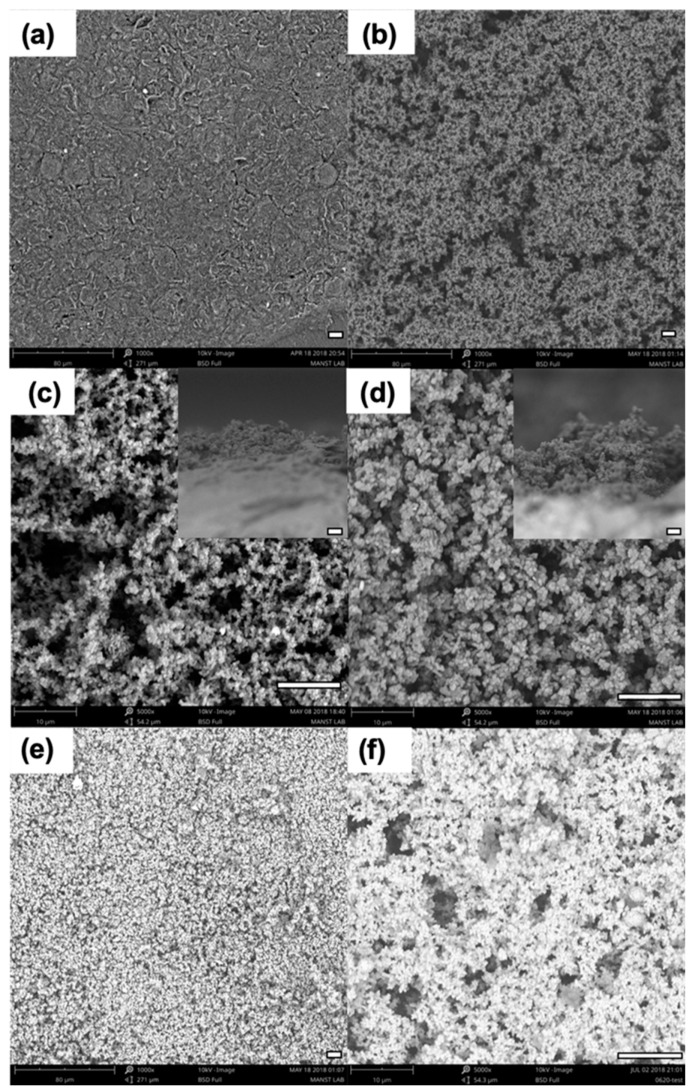
SEM images of the screen-printed carbon working electrode surface before and after modification with the PANI/AuNCs nanocomposite layer. (**a**) Bare working electrode. (**b**) After electropolymerization of PANI. The top and cross-sectional view (inset) of the PANI film obtained using (**c**) 25 and (**d**) 50 successive voltage scan cycles. After electrodeposition of AuNCs at (**e**) 1000× and (**f**) 5000× magnification. The scale bar in each image, represented by the white horizontal line, corresponds to a length of 10 µm.

**Figure 4 sensors-19-03990-f004:**
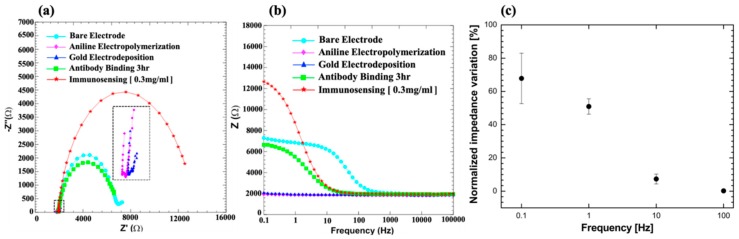
EIS characterization for each surface modification step of the SPCE and after immunosensing as represented by (**a**) Nyquist and (**b**) Bode plots. (**c**) The normalized impedance variation observed at different frequencies for immunosensing with a 300 µg/mL has-spiked sample solution.

**Figure 5 sensors-19-03990-f005:**
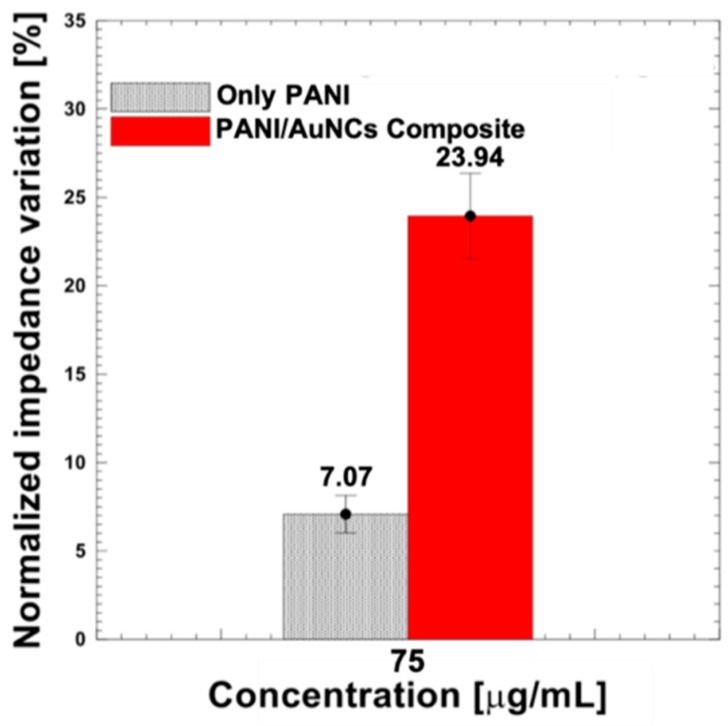
The observed immunosensor response for the same HSA concentration (75 µg/mL) when the SPCE surface was modified with PANI/AuNCs nanocomposite as compared to only PANI.

**Figure 6 sensors-19-03990-f006:**
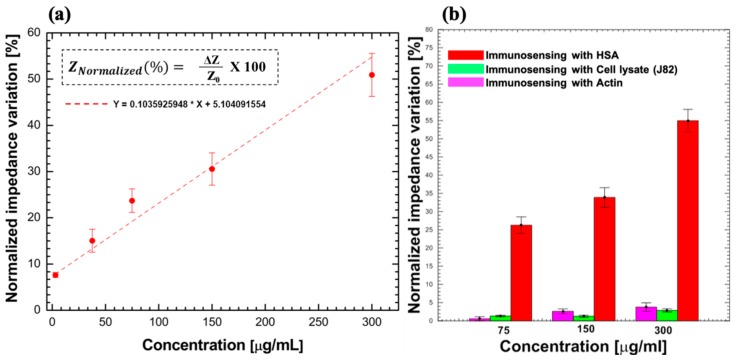
(**a**) Normalized impedance variation for different concentrations (3–300 µg/mL) of HSA in sample solution. (**b**) Interference studies to confirm specificity of immunosensor for HSA detection.

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
