# Peer review of "Impedimetric Immunosensor Utilizing Polyaniline/Gold Nanocomposite-Modified Screen-Printed Electrodes for Early Detection of Chronic Kidney Disease"

_sensors, 2019, doi:10.3390/s19183990_

Round 1
Reviewer 1 Report
Dear Author,
I think that your work is well-presented and well-written. From my perspective, your conclusions are valid but please consider the following points for the text review:
-) There are several Paragraphs 2.1 throughout the whole text;
-) Par Apparatus: have you tested other layouts for your sensors? Or maybe have you simulated the electric field distribution of other configurations, e.g., interdigitated electrodes?
-) Par. Fabrication of SCPE: please clarify how you manage to depose PANI only on the worker, i.e., have you used a custom cell to shield the pseudo-reference and counter during the PANI electrodeposition?
-) Par. Immunosensor operation: what is the molarity of the FeCN redox solution?
-) Par. EIS analysis: it seems that your definition of Rct and Cint are not used in the discussion. Do you have a complete equivalent electrical circuit for your system? A simple parallel R//C can not adequately fit your spectra. Please clarify this point in the text and show some data fit, if possible;
-) Fig. 4a: the inset is too messy, please use a separate graph for the more conductive spectra. Furthermore, they seem very noisy. Please make that figure clearer;
-) Par. EIS analysis: you state that at 1 Hz you observe a “maximum stable response”: what data are maximum stable? With respect to what? Please clarify this statement also please provide a graph to demonstrate this;
-) Par. Synergistic effect of… : please change “synergistic” in “synergic”. Furthermore, please add some data related to different target concentration to the graph in Fig. 5;
-) Fig. 6: please clarify what is Z0 and how it has been assessed. Also, please re-scale the graph along y to better fit your curve;
-) If possible, I suggest that you add some data related to the impact of human urine on your SCPE sensors, e.g., some EIS spectra without HAS, to show the validity of your technology.
Author Response
Dear Reviewer
please check our reply as attached file.
thanks
Best Regards,Cheng-Hsin Chuang Ph.D.
Associate Professor
Institute of Medical Science and Technology
National Sun Yat-sen University
70 Lienhai Rd., Kaohsiung 80424, Taiwan.
E-mail:chchuang@imst.nsysu.edu.tw Tel: +886-7-5252000 ext 5785
Mobile: 0937929830
Website:http://manstlab2015.blogspot.com/ ResearchGate: https://www.researchgate.net/profile/Cheng_Hsin_Chuang

Reviewer 2 Report
In introduction section adding some data related to Chronic Kidney diseases and deaths due to this problem can emphasize the importance of this study. An experimental table showing all the trials run, type of electrode modification used, analysis used (EIS, SEM etc) can make the study easy to interpret. The working volume for sample testing 3-300 ug/l, does it present the true result? Is it typical to use such ug/l level of volumes for the analysis? Fig 3, just based on SEM analysis can it be claimed that the coating has been done? Additional confirming test such as Raman analysis would it be required to confirm the deposition of coatings? The deposits seen on SEM could it be impurities? Does it matter with the uniformity in coating? Fig 4, using the blue color twice makes it difficult to discern the curves. Fig 4, it is mentioned the circuit elements were connected in series, it would good to provide the circuit in the Fig 4. Was the EIS analysis based on circuit fitting or was just interpreted from the figure? Also the tail portions after the semicircle seen on inset tends to disappear for immunosensing curve. Does it mean anything? For the discussion vauges terms are used for example. Page 8....It was observed that the Rct decreased significantly after eletropolymerization...It would be better to provide values of how much decrease like how many folds..Another example on Page 8.....at a frequency of 1 Hz where we observe the maximum stable response.....it would be good to see what was the value. It mentions about disposable modified electrodes. It would be automatically bring in cost factor in mind. So there should be a supporting ideas to make the point that Screen printed electrodes are comparatively cheaper.Author Response
Dear Reviewer:
please check the reply letter as attached file.
thanks!
Best Regards,Cheng-Hsin Chuang Ph.D.
Associate Professor
Institute of Medical Science and Technology
National Sun Yat-sen University
70 Lienhai Rd., Kaohsiung 80424, Taiwan.
E-mail:chchuang@imst.nsysu.edu.tw Tel: +886-7-5252000 ext 5785
Mobile: 0937929830
Website:http://manstlab2015.blogspot.com/ ResearchGate: https://www.researchgate.net/profile/Cheng_Hsin_Chuang
